META-RESEARCH ARTICLE

# "Best Paper" awards lack transparency, inclusivity, and support for Open Science

Malgorzata Lagisz[1,2]*, Joanna Rutkowska[3], Upama Aich[4], Robert M. Ross[5], Manuela S. Santana[6], Joshua Wang[7,8], Nina Trubanová[9], Matthew J. Page[10], Andrew Adrian Yu Pua[11], Yefeng Yang[1], Bawan Amin[12], April Robin Martinig[1], Adrian Barnett[13], Aswathi Surendran[14], Ju Zhang[15], David N. Borg[16], Jafsia Elisee[17], James G. Wrightson[18], Shinichi Nakagawa[1,2,19]

**1** Evolution & Ecology Research Centre and School of Biological, Earth and Environmental Sciences, University of New South Wales, Sydney, Australia, **2** Theoretical Sciences Visiting Program, Okinawa Institute of Science and Technology Graduate University, Onna, Japan, **3** Institute of Environmental Sciences, Faculty of Biology, Jagiellonian University, Cracow, Poland, **4** School of Biological Sciences, Monash University, Clayton, Australia, **5** Department of Philosophy, Macquarie University, Sydney, Australia, **6** Centre for Marine Studies, Federal University of Paraná, Pontal Do Paraná, PR, Brazil, **7** School of Clinical Sciences, Queensland University of Technology, Brisbane, Australia, **8** Mandarin Training Centre, National Taiwan Normal University, Taipei, Taiwan, **9** School of Biology and Environmental Science, University College Dublin, Dublin, Ireland, **10** Methods in Evidence Synthesis Unit, School of Public Health and Preventive Medicine, Monash University, Melbourne, Australia, **11** School of Business, Economics, and Information Systems, University of Passau, Passau, Germany, **12** Faculty of Social and Behavioural Sciences, Utrecht University, Utrecht, the Netherlands, **13** School of Public Health and Social Work, Queensland University of Technology, Brisbane, Australia, **14** School of Psychology, University of Galway, University Road, Galway, Ireland, **15** Department of Infectious Diseases and Public Health, Jockey Club College of Veterinary Medicine and Life Sciences, City University of Hong Kong, Hong Kong SAR, China, **16** School of Exercise and Nutrition Sciences, Queensland University of Technology, Brisbane, Australia, **17** African Higher Institute of Open Science and Hardware, Yaoundé, Cameroon, **18** Faculty of Medicine, University of British Columbia, Vancouver, Canada, **19** Department of Biological Sciences, University of Alberta, Biological Sciences Building, Edmonton, Canada

* m.lagisz@unsw.edu.au

**Data Availability Statement:** All data and code are openly available through a dedicated OSF project (https://osf.io/yzr7a/), Zenodo (https://doi.org/10.5281/zenodo.11215401), and GitHub repository

## Abstract

Awards can propel academic careers. They also reflect the culture and values of the scientific community. But do awards incentivize greater transparency, inclusivity, and openness in science? Our cross-disciplinary survey of 222 awards for the "best" journal articles across all 27 SCImago subject areas revealed that journals and learned societies administering such awards generally publish little detail on their procedures and criteria. Award descriptions were brief, rarely including contact details or information on the nominations pool. Nominations of underrepresented groups were not explicitly encouraged, and concepts that align with Open Science were almost absent from the assessment criteria. At the same time, 10% of awards, especially the recently established ones, tended to use article-level impact metrics. USA-affiliated researchers dominated the winner's pool (48%), while researchers from the Global South were uncommon (11%). Sixty-one percent of individual winners were men. Overall, Best Paper awards miss the global calls for greater transparency and equitable access to academic recognition. We provide concrete and implementable recommendations for scientific awards to improve the scientific recognition system and incentives for better scientific practice.

([https://github.com/mlagisz/survey_best_paper_awards](https://github.com/mlagisz/survey_best_paper_awards)).

**Funding:** Australian Research Council grant DP230101248 awarded to M.L. and S.N., Natural Sciences and Engineering Research Council of Canada (RGPIN-2019-05520) awarded to A.R.M., and John Templeton Foundation (grant ID: 62631) awarded to R.M.R. Funders had no role in designing, conducting or reporting this research.

**Competing interests:** The authors have declared that no competing interests exist.

Awards are one of the most visible forms of scientific honor [1]. They aim to signal excellence and are frequently publicized on institutional webpages, in academic newsletters, at professional conferences, and via social media. Awards can feature prominently on individual curriculum vitae (CVs) and grant applications, opening doors to jobs, promotions, and funding [2,3]. However, awards can also contribute to entrenched problems in academic evaluation if they reward papers with striking results rather than robust and transparent research. They can magnify existing disparities in science by contributing to the Matthew effect [4–6]—a cumulative effect, whereby individuals who are already advantaged gain further advantage. This feedback loop can reinforce stratification and widen the gaps for historically disadvantaged and marginalized groups in science, such as, women, carers, racial and ethnic minorities, and non-native English speakers [7–10].

Studying awards also offers an opportunity to detect and address some of the systemic issues in academia [11]. By examining awards we can gain insights into the culture and values upheld by the granting institutions and organizations and how they are changing. Specifically, examining publicly accessible materials pertaining to nomination, assessment criteria and transparency of the process can indicate whether broader social trends are reflected in academic recognition. For example, the Black Lives Matter and MeToo social movements spurred many scientific organizations and journals to post statements of support for diversity, equity, and inclusion initiatives (e.g., [12,13]). This coincided with the appearance of awards dedicated to recognizing efforts for promoting Equity Diversity and Inclusion (EDI), and also awards created exclusively for people from historically underrepresented groups, such as researchers from the Global South, racial and ethnic minorities, or women [14–17]. "Diversity awards" appear to positively contribute to the growing numbers of award winners from historically underrepresented groups [18]. Nevertheless, the members of historically underrepresented groups still receive a relatively small proportion of unrestricted research awards, especially the most valued science awards [19].

The interplay between award characteristics and awardee diversity is complex and understudied. The most prestigious research awards, such as the scientific Nobel Prizes, receive a lot of attention both from the general public and the research community. Studies of elite awards repeatedly reveal limited diversity among the winners, with unsatisfactory progress to a more balanced representation over the years, contributing to calls for more transparency and equity in research recognition [20–22]. Outside the research on elite awards, there is a growing number of discipline-specific studies, which are limited in their ability to draw broad conclusions. Nevertheless, these studies have revealed gaps of varying sizes in recognition between men and women (e.g., [15,23–30]). Few studies dissect other aspects of awardee diversity, such as race or ethnicity (e.g., [31–33]). No studies, to our knowledge, examine the characteristics of the awards per se to examine how they could contribute to such gaps (for example, by not explicitly encouraging diverse nominations or having an opaque nomination and assessment processes). Lack of transparency around the award policies and processes, in particular, can act as an insidious form of gatekeeping, which may inadvertently enable biases via personal preferences and connections [34]. Such biases likely advantage researchers from larger and better resourced groups and disadvantage researchers from historically underrepresented backgrounds.

The landscape of research awards is changing. Research awards are increasingly common across disciplines [3,35], tracking the global growth of the research community [36], the proliferation of learned societies [37], journals [38], and the pursuit of recognition for a broader range of ideas, activities, and diverse scholars. This pursuit has led to the creation of many types of awards, ranging from those recognizing broad achievements (e.g., the Association for

Psychological Science "Rising Star" awards) through to those for specific activities (e.g., communication, mentoring, application) and discrete research outputs (e.g., posters, oral presentations, books, and articles). In particular, published articles are the universal currency in most research fields and often have their own class of awards. Such awards are often termed "Best Paper" awards and are typically associated with a published journal article, which can have one or more authors. In the case of multiauthored articles, award benefits can be bestowed on one or more selected authors (usually the first or corresponding author) or shared by the whole author team. These awards can be managed by learned societies, journals, or other institutions.

There is currently limited knowledge about Best Paper awards. While academics may notice who wins such awards within their own particular disciplines, they may not have a "big picture" of the cross-disciplinary distribution of Best Paper awards—who gets recognized (e.g., in terms of gender and country of affiliation), how (award policies and procedures), and why (award criteria). There are few relevant studies, and they tend to be limited by narrow scope or small sample sizes. Nonetheless, results from discipline-specific studies are troubling. Studies in Computer Science [39] and Ecology and Evolution [11] have revealed opaque award criteria and processes. In Economics, a study has found a mismatch between subjective jury decisions and article citation impacts, similar numbers of men and women among the winners, and affiliations dominated by the United States institutions [40]. In Engineering, most winning authors were not native English speakers [41]. In Business, awardees for Women's Enterprise Development were affiliated with institutions in the United Kingdom and Canada, but originated from many regions around the globe [42]. In Ecology and Evolution, descriptions of awards targeting early- and mid-career researchers seldom consider equitability of the eligibility and assessment criteria; one positive finding was that the proportion of female winners has increased in the last 2 decades [11].

Here, we build on the above research by characterizing Best Paper awards across disciplines, with 2 main aims: (1) to examine eligibility and assessment criteria; and (2) to evaluate potential gender and affiliation country disparities in the lists of past awardees. Our findings then form the basis for recommendations on how to improve the policies, processes, and award descriptions with the goal of making publication-related awards more transparent and equitable.

## Results

We present our workflow in Fig A in S1 Text. We followed a preregistered protocol (https://osf.io/93256; see S1 Text for the minor deviations from the protocol). In brief, we first searched for awards associated with the top 100 journals from each of the 27 SCImago Subject Areas (https://www.scimagojr.com/; Fig B in S1 Text) until 10 potentially eligible awards were found or we reached the 100 journals limit, to achieve a balanced representation across disciplines. We acknowledge that the resulting snapshot of Best Paper awards may not be fully generalizable due to our search procedure and selection criteria (see Discussion). For the included awards, we extracted data on the characteristics of the awards focused on recognizing a single published journal article, because this appears to be a comparable type of research award popular across disciplines. Our data extraction covered award characteristics related to transparency, eligibility rules, and assessment criteria. For awards specifically highlighting individual article authors (i.e., having a main winner/s rather than the whole authorship team being recognized equally), we inferred the winning author's gender and country of affiliation, for winners from the years 2001 to 2022. We present overall results (across all Subject Areas) in the main text and we include equivalent results by Subject Area in S1 Text.

## Awards—General characteristics

Across all 27 Subject Areas, we included 222 unique awards (4 to 10 per Subject Area; Fig 1A). For 11 of the included awards (5%), we did not find any award description text (Fig 1B), but we still were able to use information on past award winners. The median number of words in the available award descriptions was 123 (Fig 1C; word counts by Subject Area in Fig C in

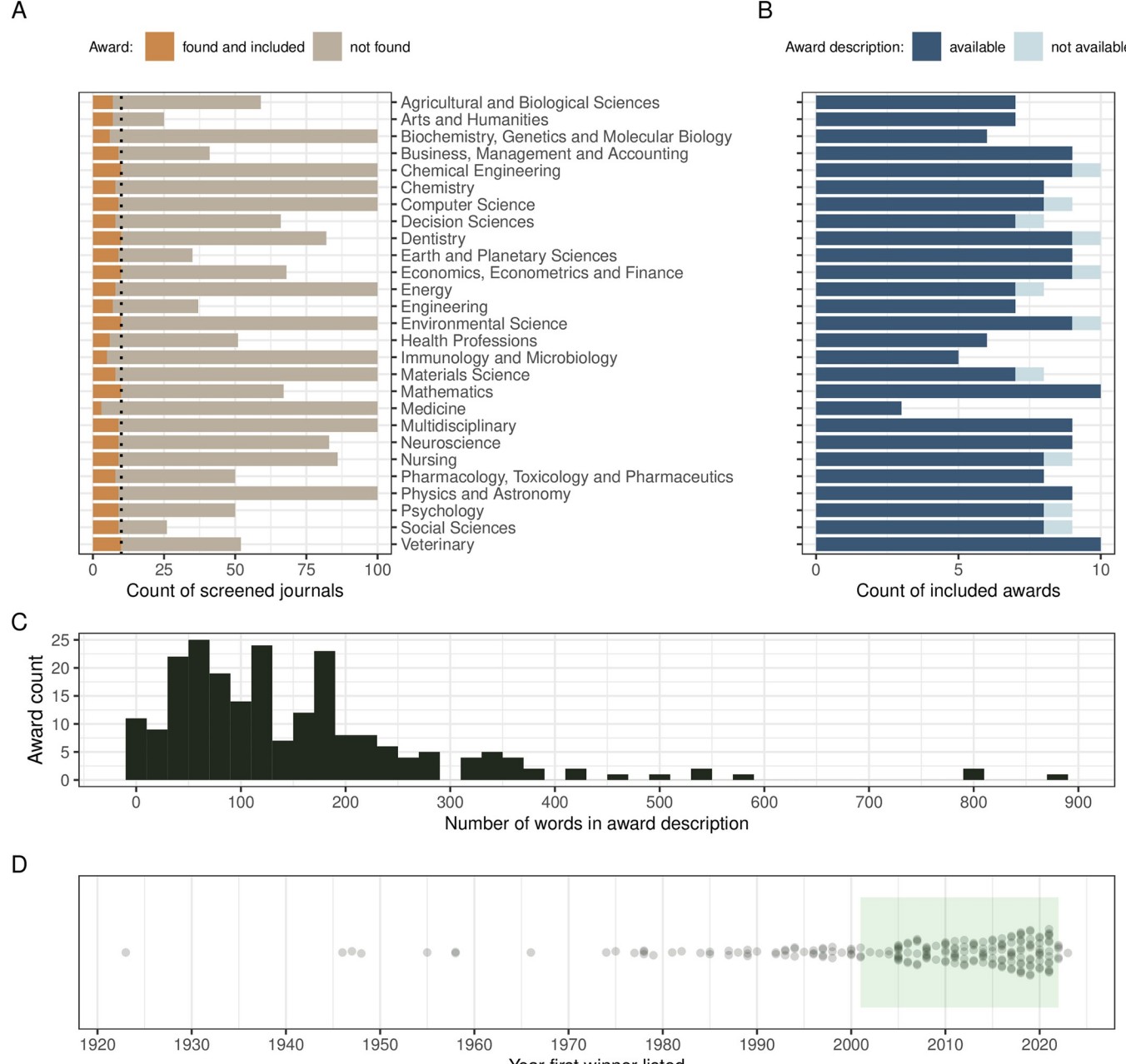

**Fig 1. Summary of award searching effort and general award characteristics.** (A) Counts of the journals screened from the 27 SCImago Subject Areas rankings and counts of the eligible awards included for data extraction per Subject Area. (B) Counts of the included awards with and without award description, per Subject Area. (C) Distribution of numbers of words per award description. (D) Distribution of the year of the earliest winner found for each included award (green rectangle marks the period from which information on individual winners was extracted). Vertical dotted line in panel A indicates a cut-off of 10 awards per SCImago Subject Area. The data and code needed to generate this figure can be found in https://doi.org/10.5281/zenodo.12465262.

S1 Text). The oldest included award had its first winner listed in 1923; however, 48% of the awards in our data set were established after the year 2010 (108; Fig 1D and Fig D in S1 Text). Of the included awards, 176 (79%) were associated with specific journals (Fig 2A), 144 (65%) with learned societies, and 39 (18%) also mentioned some other organizations being involved (usually journal publishers providing financial or other support: e.g., Elsevier—15 awards, Wiley—6; many awards had more than 1 type of awarding body, so that percentages do not add up to 100%).

## Awards—Eligibility and inclusivity

In terms of eligibility, 179 (81%) of the awards did not explicitly target specific career stages. The remaining awards mostly targeted early-career authors, students or different combinations of early career stages (Fig 2B). In terms of author focus, 66 (30%) of the awards were focused on selecting and distinguishing a specific author of the winning article ("individual awards"), in contrast to the remaining awards where the whole authorship team were announced as winners ("team awards"; Fig 3A and Fig E in S1 Text). For the team awards, eligibility was not based on the characteristics of individual authors and thus restrictions related to career stage were not applicable. For the individual awards, this was also true for 28 (42%) awards that did not target a specific career stage. For the individual awards that set some career-related eligibility time limits (usually expressed in years since PhD), 21 (81% out of 26) did not mention whether eligibility can be extended (e.g., in case of career breaks) and only 5 did (19%; Fig 3A and Fig F in S1 Text). In terms of encouraging historically underrepresented and marginalized scholars to apply for the awards, only 2 awards (1%) provided inclusivity statements (Fig G in S1 Text).

## Awards—Transparency

Over half of the awards (114; 54%) reported who the award assessors are (either listing specific names or mentioning journal editors; Fig H in S1 Text). Integrity issues related to potential

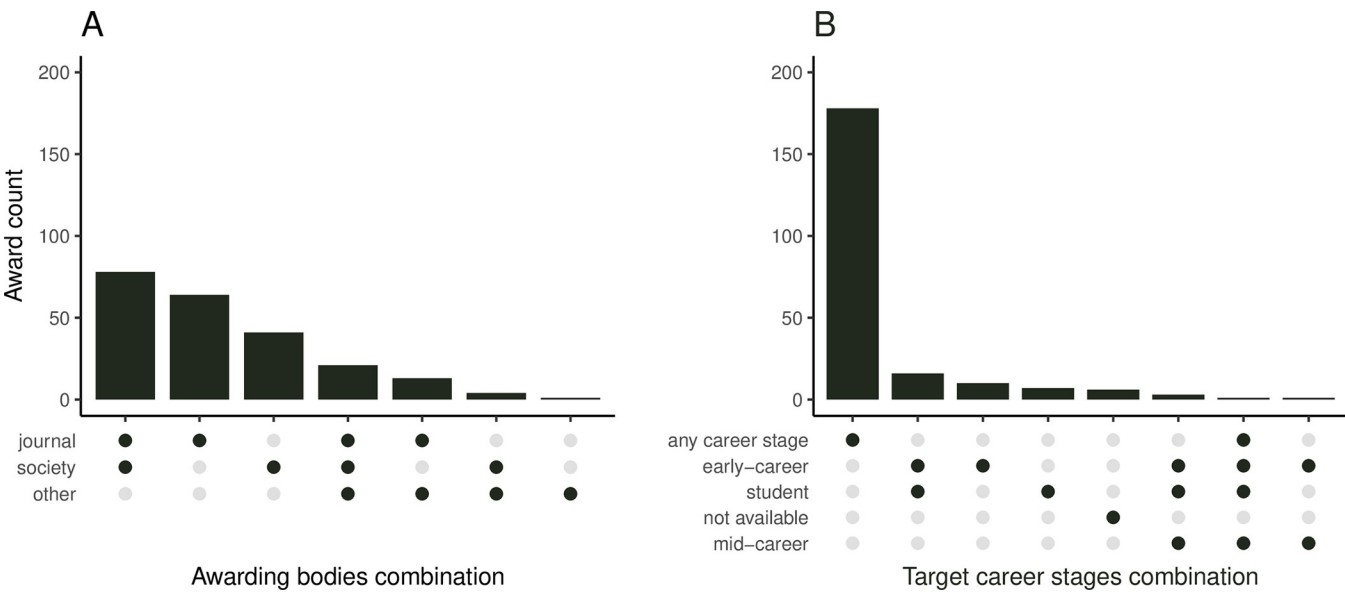

**Fig 2. Summary of included awards.** (A) Awarding bodies. (B) Award target researcher career stage. "Not available" stands for cases with no award description. The dark-colored dots in the bottom part of the graphs indicate which combination of the awarding bodies (A) or target career stages (B) a given vertical bar above represents (e.g., first bar in panel A shows the count of awards that are associated with both a specific journal and a learned society, and the second bar is for awards associated with a journal only). Awarding bodies and target career stages are shown in the order of their decreasing frequency. The data and code needed to generate this figure can be found in https://doi.org/10.5281/zenodo.12465262.

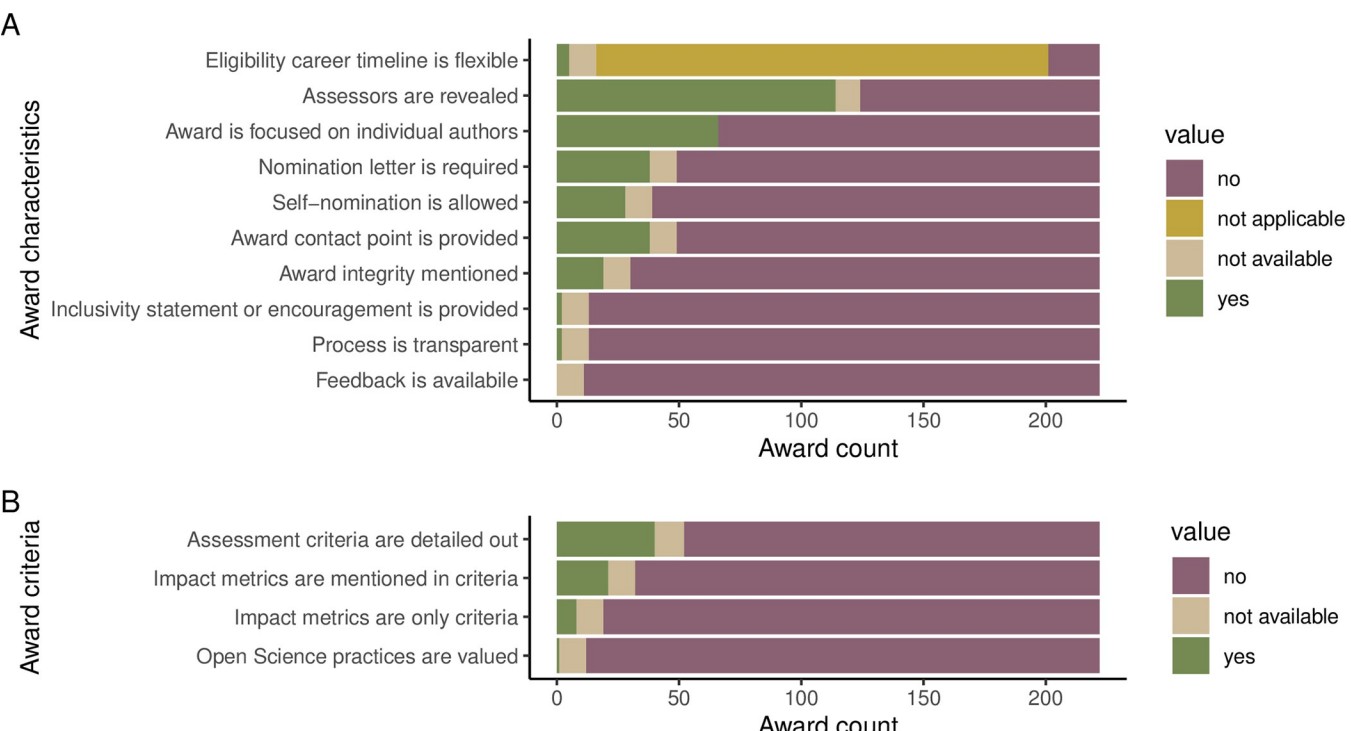

**Fig 3. Summary of characteristics of included awards, derived from award descriptions.** (A) General characteristics related to accessibility, inclusivity, and transparency. (B) Assessment criteria for nominated articles. "Not available" stands for cases with no award description. The data and code needed to generate this figure can be found in https://doi.org/10.5281/zenodo.12465262.

conflicts of interest (e.g., whether articles co-authored by the assessors can be nominated) were mentioned in 19 award descriptions (9%). Only 2 awards provided gender or affiliation information about the nominated authors alongside the winning articles ("Process transparency" in Fig 3A and Fig I in S1 Text). Self-nominations were explicitly allowed by 28 awards (13%; Fig J in S1 Text), while 38 asked for a nomination letter from the nominating person (18%; Fig K in S1 Text), and 10 mentioned both (5%). None offered potential feedback to the nominees (Fig L in S1 Text), and 38 (18%) of the award descriptions or pages included some contact details that could be used to enquire about the award (Fig M in S1 Text).

## Awards—Assessment criteria

Most award descriptions were not only brief but also vague—only 40 awards had assessment criteria that were not restricted to generic statements like "excellent research" (19% of awards with descriptions; Fig 3B and Fig N in S1 Text; [43]). From the 21 awards (10%; Fig O in S1 Text) that mentioned article impact metrics (counts of citations or downloads) in their assessment criteria descriptions, 8 used such metrics as their only award criteria (4%; Fig P in S1 Text; more common in recently established awards: Fig Z2 in S1 Text). Practices relevant to Open Science (such as transparency, reproducibility, and robustness) were mentioned only in the criteria of a single award (0.5%; Review of Research Award by the American Educational Research Association; Fig Q in S1 Text) explicitly emphasizing "transparency of the methods." Additional text mining of the award descriptions revealed that words such as "best," "outstanding," "original," and "impactful" frequently appear in award descriptions, but words related to reproducibility and robustness are rarely (if ever) mentioned (Fig 4).

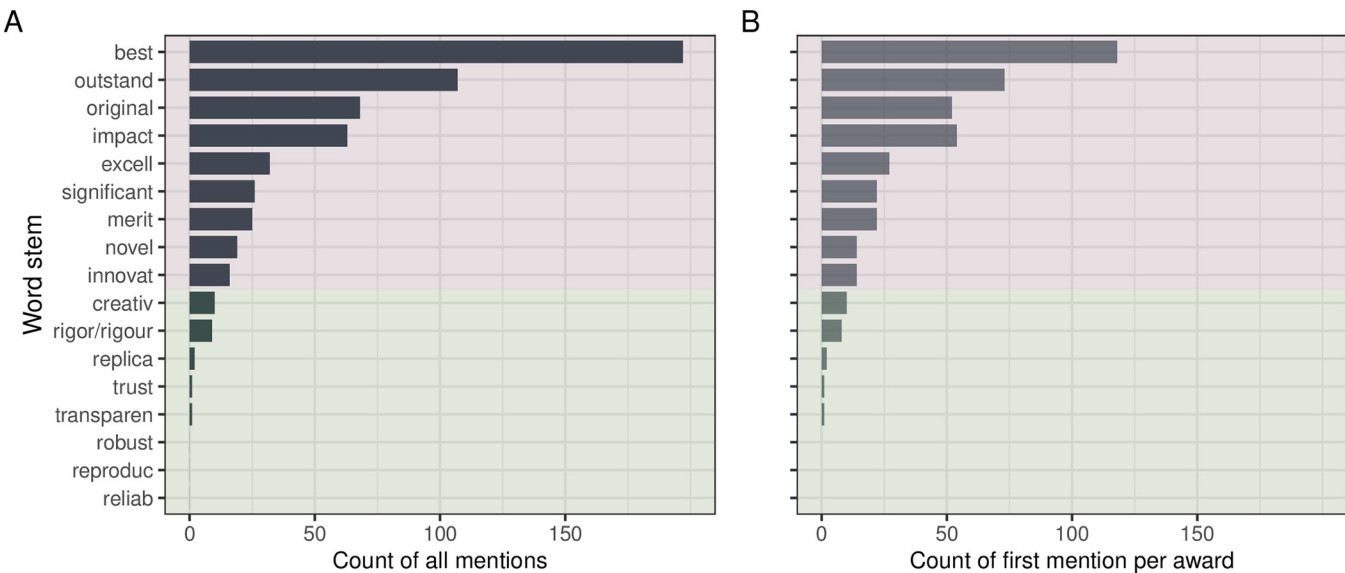

**Fig 4. Frequencies of selected words (stemmed) in award descriptions.** (A) Counting all mentions in all awards descriptions. (B) Counting only the first mention per award description. The pink shaded area encompasses terms related to generic excellence and other buzzwords; the green area encompasses terms related to scientific transparency and robustness. The data and code needed to generate this figure can be found in https://doi.org/10.5281/zenodo.12465262.

### Past winners—Gender and affiliation country

For the years 2001 to 2022, individual awardee data were available for 1,079 past winners representing 61 unique awards across 19 Subject Areas (range of 0 to 8 awards per Subject Areas, with a median of 3, Fig R in S1 Text; 0 to 152 winners per Subject Areas, with a median of 42; Fig S in S1 Text; annual trends in Figs T and U in S1 Text). Overall, men won more individual awards than women (61% and 39%, respectively; Fig V in S1 Text) with the gender ratio stable across decades (60.6% of men in 2001 to 2010, 60.7% of men in 2011 to 2020, 60.8% in 2021 to 2022; Fig 5A and 5B; annual trends and by Subject Area in Figs W–A2 in S1 Text).

We retrieved affiliation information for 1,029 winners representing a total of 44 countries (Figs B2–F2 in S1 Text), as reported on award pages, announcements, or on the winning articles. USA-affiliated authors received the most awards across decades (Fig 5C), but their share of the awards declined over time (Fig 5D and Fig G2 in S1 Text). This was matched by an increasing proportion of winners with other affiliations, especially China, Canada, France, Germany, Italy, and by the increasing number of unique countries represented (26 in the decade 2001–2010, 39 in the decade 2011–2020, and 29 in just 2 years 2021–2022). Still, the most frequent affiliation institutions were all located in the USA, with the top 3 being: University of Washington, University of California San Diego, and University of California Los Angeles (13 winners each), followed by another 10 US universities (Fig K2 in S1 Text). Overall, USA-affiliated researchers received 48% of the awards, which is more than their 15% annual contribution to the SCImago-listed articles in 2021 (Fig H2 in S1 Text). In contrast, countries outside the top 10 won 19% of the awards while contributing 44% of the articles in 2021 (Fig H2 in S1 Text). Further, there were only 111 winners (11%) affiliated with institutions in the Global South countries, representing only 12 countries (mostly China—61 winners; Fig B2 in S1 Text; by decade and gender Fig I2 in S1 Text). The most frequent first names among the past winners were: Sarah (12), David (10), Christopher (8), Michael (8), Jonathan (7), Richard (7), Anne (7), Adam (6), Andrew (6), Benjamin (6), Brian (6), Daniel (6), Eric (6), James (6), Philipp (6), and William (6) (Fig J2 in S1 Text).

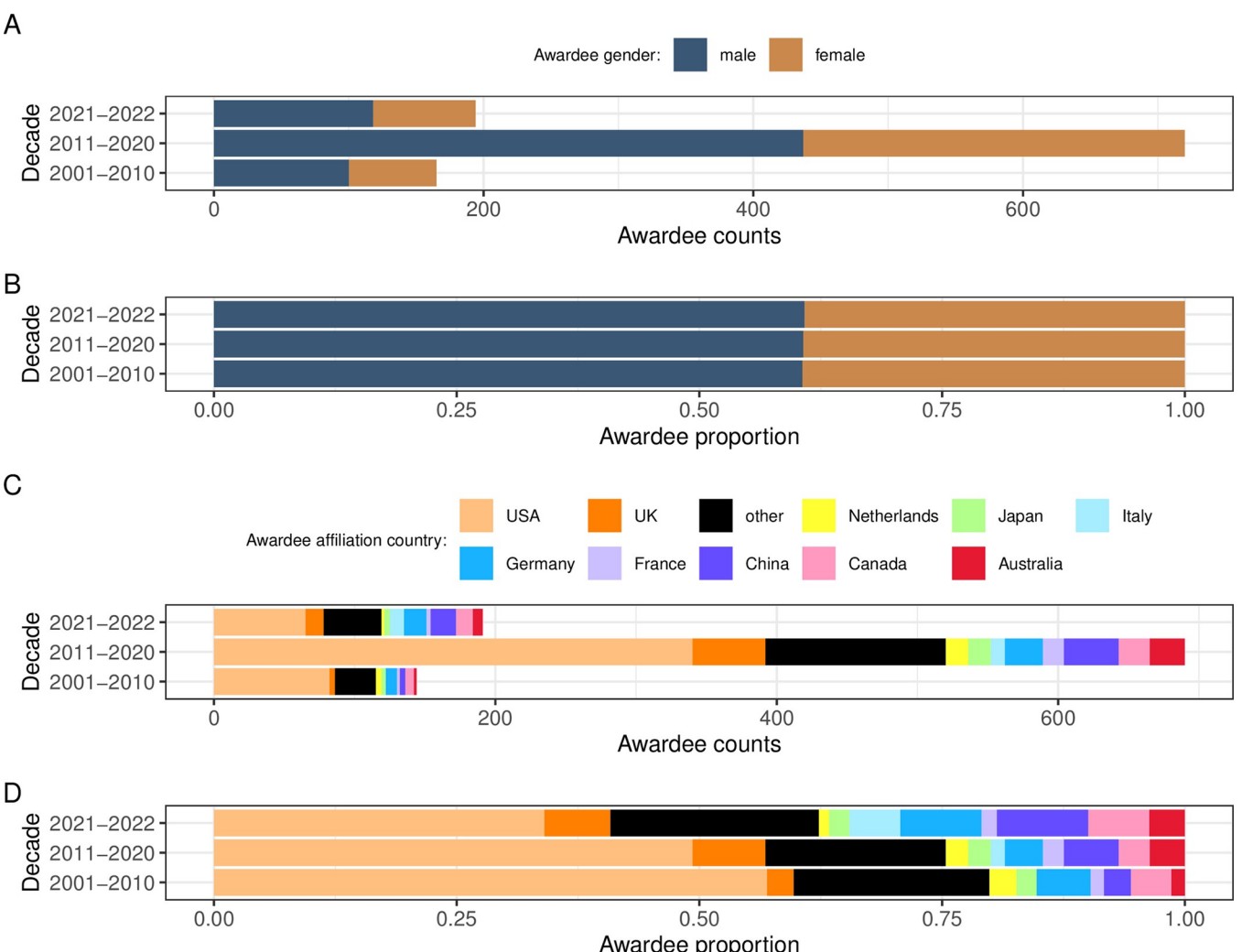

**Fig 5. Gender and affiliation country of the past award winners across years 2001 to 2023.** (A) Counts by gender per decade. (B) Proportion by gender per decade. (C) Counts by affiliation country per decade. (D) Proportion by affiliation country per decade. Gender was assigned to individual past award winners using available pronouns, photos, and first names. Only top 10 most frequent affiliation countries are shown, with the remaining 33 countries aggregated into the "other" category. Color-shaded map pots of all affiliation countries are provided as Figs F2 and G2 in S1 Text. The data and code needed to generate this figure can be found in https://doi.org/10.5281/zenodo.12465262.

## Cash and other perks for the winners

In terms of the financial benefits, 112 (50%) of the awards descriptions mentioned a monetary prize for the winning authors (Fig 6A and Fig L2 in S1 Text). The majority of the cash prizes are less than 2,500 USD (median = 1,086 USD; Fig 6B), but there are also 4 valued at 10,000 to 25,000 USD (Fig M2 in S1 Text). Exploratory text mining revealed award benefits other than cash (Figs N2 and O2 in S1 Text). The tangible benefits (that go beyond the symbolic certificate and/or plaque), include travel grants, invited talks, waivers of conference fees, membership, subscription fees, or publication fees. Finally, it is increasingly common for individual award winners to have their research profile and/or photographs posted on a public award webpage or included in award announcement (Fig 7A and 7B).

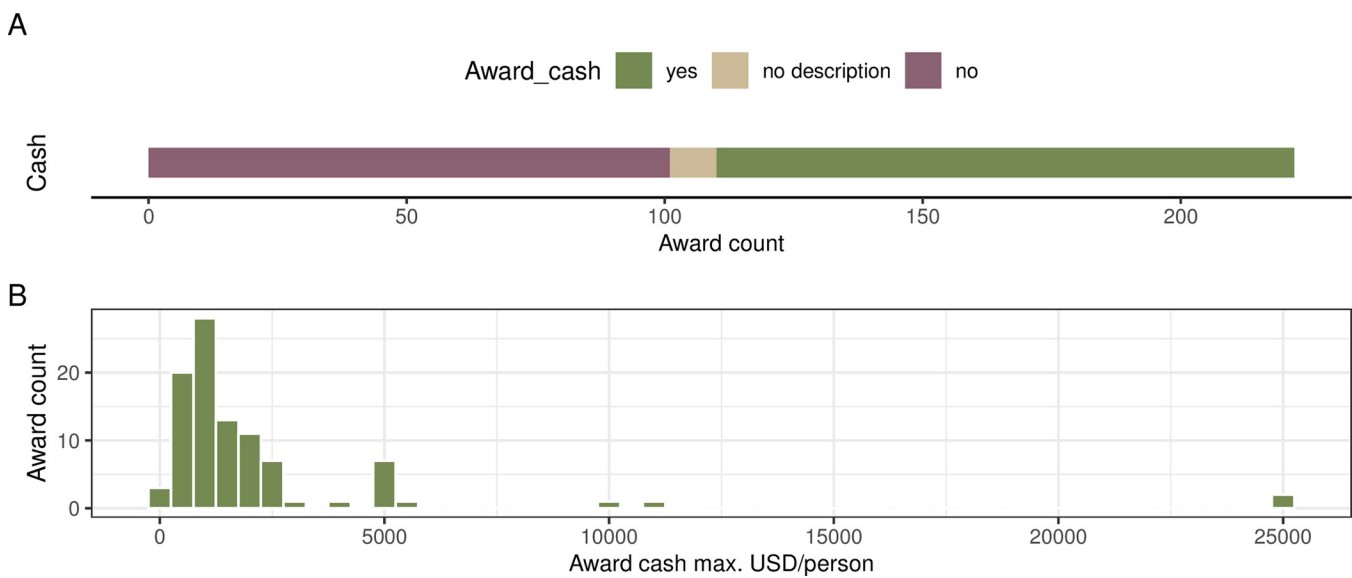

**Fig 6. Award monetary prizes and other perks.** (A) Information on whether the award description mentioned cash prizes. (B) Disclosed amounts of cash prizes for 97 awards (2023 values), recalculated into US Dollars. The data and code needed to generate this figure can be found in https://doi.org/10.5281/zenodo.12465262.

## Discussion

Our investigation of Best Paper awards across all 27 SCImago Subject Areas uncovered a general lack of transparency in publicly available descriptions of the awards. Nonetheless, we were able to identify patterns in what was reported in terms of the award process, criteria, benefits, and winners, including historical trends in terms of gender and affiliation. Below, we summarize our specific findings and then provide suggestions on improvements and future research.

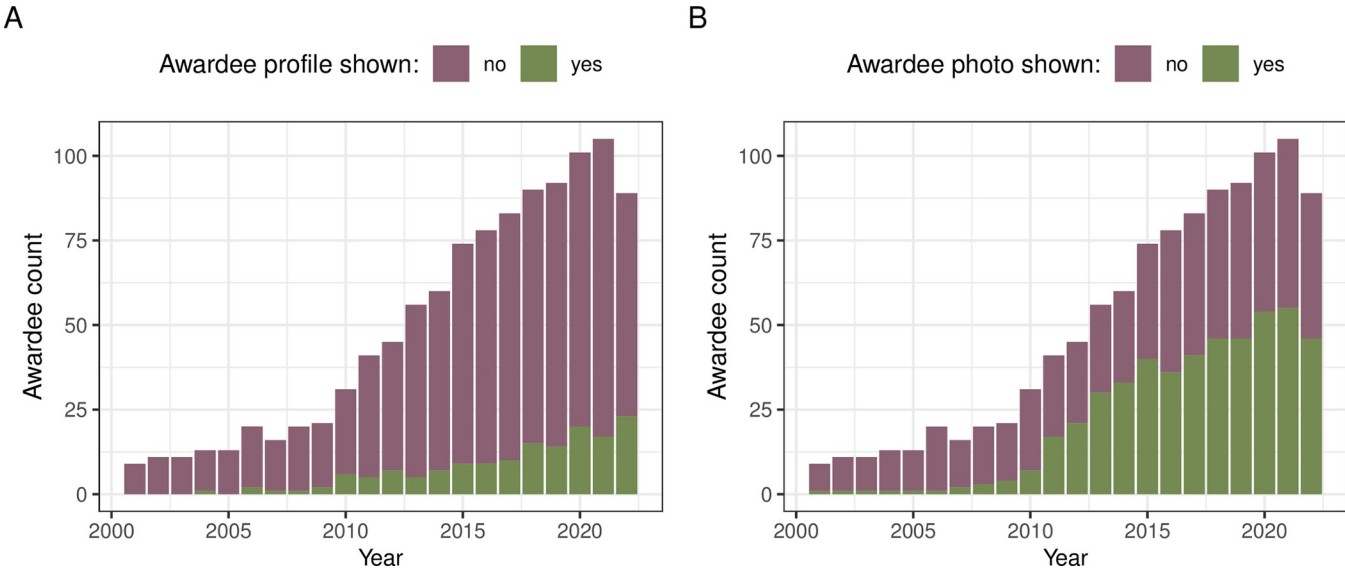

**Fig 7. Recognition of individual award winners via official award webpage postings of the winners' profiles and photos.** (A) Annual counts of winners with personal profiles posted on an official award webpage or announcement. (B) Annual counts of winners with personal photos posted on an official award webpage or announcement. The data and code needed to generate this figure can be found in https://doi.org/10.5281/zenodo.12465262.

Our results indicate that Best Paper awards usually are not restricted to any career stage and discipline, and are growing in number (award proliferation has been already noted in 2008 [44]). However, in our project, characterizing award eligibility and assessment criteria was hindered by a lack of award descriptions (5% of eligible awards had no descriptions) and by the available descriptions being brief (median 123 words). Assessment criteria were usually vague, while excellence-related buzzwords were frequent. Words related to comprehensive reporting, rigor, and reliability were almost nonexistent, indicating that related qualities may not be considered when evaluating published works. Concerningly, award descriptions may simply mirror the trend for the increasingly sensationalized language observed in the abstracts of articles and grant applications [45–47].

Only 1% of award descriptions included Equity, Diversity, and Inclusion statements or encouragement for minority and historically marginalized researchers to apply or be nominated. Among individual awards with time-bound eligibility restrictions, few offered eligibility extensions for career interruptions (8%), which are known to disproportionately affect underrepresented and underprivileged individuals [48]. While 54% of the award descriptions had statements about who will be assessing candidate articles, they mostly merely stated that journal editors will be selecting the winning papers rather that listing assessors' names. Disclosures of the diversity of the past nomination pools were also absent, making it difficult to examine whether candidate diversity is reflected in the diversity of the winners. All the above observations appear to be generally in line with the results of earlier work [11], characterizing a small set of early-career awards in ecology and evolution. We found that the information on how potential conflicts of interest will be handled is seldom provided (9% of Best Paper awards), and a minority of award descriptions (18%) include contact details for enquiries, further reducing transparency.

Do Best Papers awards recognize research quality or follow the path of least resistance? Despite the generally poor transparency, we found that 10% of awards descriptions mentioned article impact metrics (counts of citations or downloads), and 4% relied exclusively on such indicators (which is likely an underestimate as we also found some "Impact awards" which are awards based solely on number of citations or downloads, and did not fit our inclusion criteria). Use of impact indices gives the award criteria a veneer of objectivity and simplifies the assessment process [49]. However, their increasing use in selection and assessment of best articles is also concerning. It counters the movement towards abandoning such indices in research assessment due to their inherent biases (e.g., [50]). Notably, initiatives such as the San Francisco Declaration on Research Assessment (DORA) and Coalition for Advancing Research Assessment (CoARA) champion moving away from simplistic quantitative indicators and emphasize importance of transparent criteria and processes for research assessment [51]. It remains to be seen how such initiatives will influence Best Paper awards.

Among our findings, the lack of mentions (in all but one case) of the importance of adherence to Open Science practices shows clearly that such practices are not valued (or at least not articulated publicly as valuable or required). In contrast, the fact that 70% of the surveyed awards were not focused on individual authors could be interpreted as giving recognition to the team work, following the calls for more equitable distribution of the credits in science [52]. At the same time, it removes the need for assessing the eligibility and contributions of individual authors, making administration easier. On the flip side, individual-focused awards, especially these targeting early career researchers, create an opportunity for them to gain visibility in the research community [2]. The visibility comes not only from public award announcements (on websites, editorials, and/or during conferences), but also from invited conference talks, and research profiles and photos published online.

Additional perks of Best Paper awards may include membership and publication fee waivers, travel grants, and cash. Even if the median monetary value of Best Paper awards (1,086 USD) cannot compare with more elite awards [53], they can be a game changer for the early-career, underprivileged, and underfunded researchers, especially those from the Global South countries or underfunded labs. They not only provide a financial boost, but also demonstrate a record of attracting external funding and contribute to an invaluable sense of achievement and belonging, especially for minority and underrepresented researchers [54].

Our survey of awards included quantifying gender and affiliation country patterns for lists of past awardees to reveal how Best Paper awards may potentially contribute to the systemic disparities in career progression of historically marginalized and underrepresented groups in research. We found that, overall, 61% of the winners were men, and that this proportion did not shift over the last 22 years (Fig N2 in S1 Text). Lagisz and colleagues [11] reported increasing parity in awards for early-career researchers in ecology and evolution, but in their analyses encompassing a period of over 50 years the dominance of men among winners was clear before the year 2000 and almost balanced-out in the decades after 2000. Numerous studies reported gender biases in expert evaluation of grant proposals, with varying time-trends (see, [55]). In terms of country of affiliation, USA-affiliated researchers dominate the lists of awardees, but recently an increasing share of the awards is taken by other countries, still mostly from the Global North and China (the only highly visible Global South country in our data). Noticing, valuing, and promoting the work by the researchers from the Global South (as well as other underrepresented groups, such as women) would require changes in how Best Paper awards are defined, managed, and presented, as outlined below.

Our findings have implications for the broader scientific community (using the awards in researcher assessments; [34]), for the awarding bodies (designing and publicizing the awards), and for individual researchers (applying or nominating for the awards).

Recommendations for the research community include asking whether we should use Best Paper awards in researcher assessments. This is because it is often unclear if these awards actually reflect the quality of the science presented in the winning papers (not the focus of our study) or just striking findings (as indicated by the language in award descriptions), and whether we can trust the oblique process that led to the winner selection. Thus, increasing transparency of would be a first step toward building community trust [56], revealing their true value and acknowledging limitations of Best Paper awards.

Table 1 presents our recommendations for the awarding bodies (institutions and groups that design, fund, manage, and publicize the awards). The table is structured following the SPACE (Standards for scholarship; Process mechanics and policies; Accountability; Culture within institutions; and Evaluative and iterative feedback) rubric of 5 institutional capabilities that should be considered when reforming research assessment [57,58]. From the SPACE rubric, we used the Foundation column only, which is the first state of readiness for reform, encompassing core definitions and shared clarity of purpose.

Recommendations for individual researchers potentially applying or nominating for the awards are focused on proactive communication with the awarding bodies. If the current award description is vague or unclear—ask for clarifications. If no specific contact details are provided—ask the awarding body for the relevant contact details. Also ask if they could share statistics on past nominations and winners—numbers and demographic summary information. This way you not only learn how to apply, and what your chances may be, but you also apply pressure on improving the transparency and equitability of the award itself.

Our data have 5 main limitations. First, our process for identifying relevant awards for inclusion was stratified (focusing on top ranked journals) and thus not exhaustively comprehensive or fully random which may reduce the generalizability of our findings. We attempted

**Table 1. Recommendations for awarding bodies establishing new and managing existing Best Paper awards across disciplines.** These recommendations could also be helpful for improving transparency and equity when considering including Best Paper awards in research assessment. The table is based on SPACE rubric [57,58], but the specific recommendations are largely derived from the award survey results.

| Institutional capabilities | Proposed starting points for systemic changes (Foundation) |
|---|---|
| STANDARDS FOR SCHOLARSHIP: *How are new definitions of "quality scholarship" formulated and applied*? | ALIGNMENT on values and goals: <br> • Clearly define what "Best" stands for, without relying on buzzwords (e.g., excellence, novelty, significance). <br> • Explicitly design and explain standards for eligibility and assessment, including specific definitions and standards of article quality. <br> • Align standards with the institutional mission and values. <br> • Value Open Science practices, such as study registration, preprints, public sharing (meta-)data, and code (if applicable), as indicators of transparency and reliability of research. <br> • Focus on increasing equity and support for traditionally underrepresented and minoritized groups. <br> • Work on diagnosing the nature of inequities that exist in the current (or traditional) policies and procedures. |
| PROCESS MECHANICS AND POLICIES: *How are new practices incorporated into review structures, processes, and institutional policies*? | DEBIASING deliberative judgments: <br> • Foster assessment of nominations that is consistent and structured to minimize potential biases. <br> • Require nominators and assessors to disclose potential conflicts of interests and establish processes to mitigate such conflicts. <br> • Go beyond traditional ways of evaluation by considering additional evaluative contexts (e.g., access to resources, local policies) to increase equitability of the research assessment. <br> • Consider splitting the award, randomly picking the winner, or handling multiple awards if there is no clear single winning article or individual. <br> • Be deliberate in justifying the focus of the award on the article (the authorship team) or on an individual author (e.g., first or corresponding author, early-career researcher), and tailor your processes and policies accordingly. <br> • Value diverse and inclusive research teams (e.g., collaborating outside the immediate circle of colleagues, gender-balanced teams). <br> • Use the author contributorship statements (and request such statements) in assessments for individual-focused awards. |
| ACCOUNTABILITY: *How are individuals and institutions held liable for executing on new assessment practices*? | TRANSPARENCY and clarity of goals: <br> • Publicly post documents defining all policies and procedures to reduce confusion, build trust, and lead by example on transparency. <br> • Create a public record of numbers of nominations each year—the more competitive the award, the greater the honor of winning it. <br> • For individual-focused awards, collect and post a summary of demographic characteristics of the nomination pool. <br> • Provide contact details (e.g., email/online contact form) for enquiring about the award (e.g., to the chair of the award committee or award secretary). <br> • Disclose all the financial and nonfinancial benefits for the winners, including cash amounts. |

(*Continued*)

**Table 1.** (Continued)

| Institutional capabilities | Proposed starting points for systemic changes (Foundation) |
|---|---|
| CULTURE WITHIN INSTITUTIONS:<br>*How are assessment practices perceived and adopted both within and outside of formal evaluation activities*? | INCLUSION and access:<br>• Ensure that diverse types of individuals are involved in both (re)designing the award and assessing award nominations.<br>• If nominations are required, explicitly encourage diverse nominations.<br>• Advertise award nominations broadly to increase the chances of that representation of minoritized applicants to meet or exceed your equity goals.<br>• Allow self-nominations and do not place weight on the nominator's identity or prestige (consider anonymizing).<br>• Post the full profile (including bio and photo) of the individual winners on the award page. Consider doing this in more than 1 language for greater accessibility. |
| EVALUATIVE AND ITERATIVE FEEDBACK:<br>*How are intervention outcomes and progress toward institutional values captured and continually improved upon*? | ARTICULATION of diverse indicators:<br>• Track progress towards more transparent and equitable awards using a range of indicators, such as diversity of assessors, nominations, runners-ups, and winners.<br>• Seek formal and structured feedback about the award policies and procedures from the assessors, nominators, nominees, winners, and your broader research community.<br>• Use feedback to improve award policies, processes, and documentation. |

to create (following predefined procedure) a snapshot of Best Paper awards from across all Subject Areas. However, Best Paper awards appeared to be more widespread in some and less popular in the other Subject Areas, resulting in uneven representation. The second, related, complication was that many journals linked to the awards are classified to more than one Subject Area—just among top 100 journals per Subject Area, 24% appear in more than 1 Subject Area (and across all listed journals this number is 40%; some journals appear as misclassified). Third, 48% of the awards appeared to be established after 2010, with new awards being established every year. Thus, we often had only a few years of available data on the past winners (some older data might be missing), potentially influencing our analyses of the historical trends. Fourth, when assigning gender to the past winners, in 58% of cases we only had a person's first name as the basis of our coding (we also relied on pronouns and photos when available). We acknowledge that the person's publishing name (and officially used images or pronouns) may not accurately reflect the person's gender self-identification [59]. For the same reason, we did not attempt to code race/ethnicity ancestry and only extracted country and institution from affiliation information. Other dimensions of diversity that cannot be surveyed from publicly available information include disability, linguistic, socioeconomic, educational background, etc. We note that the country of affiliation only represents the place of work at the time of award, which may be a poor proxy for a person's country of origin. Nonetheless, affiliation information tends to reflect the prestige and resources available to the researcher during their research work and how the world sees research success of a country [60]. Finally, our analyses of the winner's country of affiliation representation do not directly control for the number of papers published by researchers in each country. However, we provide relevant context in S1 Text. We caution against drawing simplistic conclusions from our work and we next point out new avenues for future research.

Further research on awards is needed to provide evidence and guide changes in relevant policies and practices—ultimately making them more transparent and equitable. Here, we

propose 3 future directions for such research, complementing and building on our work. First, because our data set included a limited number of Best Paper awards per discipline (as represented by SCImago Subject Areas), future work should investigate other types of awards and consider more comprehensive analyses within individual disciplines. Second, research on the diversity of the past winners could be extended by collecting the data directly from the award winners and nominees (e.g., via online surveys). Such mode of data collection would allow award winners to self-identify their gender, geographic origin, and other characteristics (e.g., racial/ethnic, disability, linguistic, socioeconomic, educational). The surveys could also gather perceptions and experiences with the award process and how winning a Best Paper award influenced the winner's career. Surveying award committees, journal editors, and the general research community could reveal their attitudes towards these awards, winning authors, or their research. Such multifaceted studies may reveal sentiments and experiences that cannot be derived from the publicly available award information alone, but are essential for interpreting such data and driving systemic changes [61]. Finally, we can turn our attention to the winning articles. While the quality of published research is hard to define and measure, it is possible to assess whether winning articles are more likely than other papers to comply with Open Science practices. By this, we do not mean that the papers have to be available under Gold Open Access—which has been implicated in driving growing inequity in research [62–64]. Instead, future research on the winning articles should have a clear focus on assessing practices that are free to do, such as publicly posting pre/post-prints, study registration, open (meta)data, and code, when applicable [65–68].

## Conclusions and further action

Our cross-disciplinary survey fills the gap in the studies of academic recognition via research awards. It revealed a general lack of transparency in Best Paper awards across disciplines, potentially undermining trust in the awards themselves. Ubiquity of brief and unclear publicly available award descriptions calls for a deep rethink and rehaul. Such changes should be in line with the values held and hailed by the scientific organizations and communities, including Open Science practices. This is because Open Science practices (such as study registration, preprints, public sharing of data and code, when applicable) do not appear to be currently valued by the bodies administering the awards, lagging behind the broader social movement towards more transparent and reliable research. We also encourage these bodies to evaluate potential biases in the award nominations and among the winners and how their awards may contribute to systemic biases in academia. Rectifying such potential biases is our collective responsibility. For the journals and societies that publicly state their commitment to Equity, Diversity, and Inclusion, as well Open Science, the disconnect between the current state of Best Paper awards remains inexplicable.

## Methods

### Protocol

A detailed plan of this study has been preregistered on the Open Science Framework (https://osf.io/93256), with contributions from ML, SN, YY, UA, BA, JR, MJP, JB, KK, AAYP, NT, and RMR. S1 Text file provides additional details of our search, screening, and data extraction process, and lists all deviations from the planned procedures.

### Search and award screening

"Best Paper" awards are by definition given for a single published academic article and are usually associated with journals or learned societies. To investigate characteristics of such awards

across all research disciplines, we sampled evenly across all SCImago Subject Areas (journals are classified into 27 major thematic categories according to Scopus Classification [69]). We also decided to focus on awards that carry the most prestige, have high visibility, and value. To collate such a sample of awards, we conducted a systematic-like search based on top-ranking journals, and associated learned societies, based on the assumption that awards associated with highly ranked journals will be more prestigious.

We first retrieved journal rankings lists for each of the 27 SCImago Subject Areas (https://www.scimagojr.com; openly available dataset for year 2021). We planned to screen consecutively the top 100 journals from each Subject Area list. For each screened journal, we first searched its website, and then also a website and announcements of any directly associated learned societies, to find any mentions of relevant awards. We stopped screening when we located 10 potentially relevant awards per Subject Area or when we reached the threshold of 100 screened journals per Subject Area. Award eligibility was cross-checked by a second reviewer against our selection criteria, with disagreements resolved by discussion. We only included awards for a single published research contribution in a format of a research article. We did not consider awards that were discontinued and awards that were restricted to applicants from underrepresented groups (e.g., women-only/minorities-only awards). We then extracted data from the awards that met our selection criteria.

## Data extractions

Data extraction was based on information gathered from the websites (e.g., journals, societies, publishers) or other publicly available documents (e.g., instructions for applicants). The extracted data concerned information on the granting journal/publisher/society, award eligibility criteria, award assessment criteria, benefits for the winners, and identities of the past winners from the years 2001 to 2022.

The extracted variables were designed to capture award description transparency, target nominee career stages, potential barriers to diverse award nominations, and mentions of Open Science practices or values in the award descriptions. We assigned gender to past awardees by using their pronouns, photos, and first names available on award websites, personal, or professional websites. We also manually located information on awardees primary institutional affiliations (country and institution) from award websites or awarded articles. For the key data items, we recorded relevant quotes from award descriptions in the dedicated comment variables, as available. We archived award descriptions from the websites as pdf files. We followed an agreed protocol and instructions for screening and data extraction process based on custom-built Google forms to assure high consistency of all extracted data. Extracted data were cross-checked by a second reviewer. Detailed description of all collected data (meta-data) is provided in Tables A–D in S1 Text and includes information on author contributions to all stages of the data extraction and checking process. ML, SN, YY, UA, BA, JR, AB, MJP, JB, AAYP, NT, RMR, JW, JE, DB, ARM, MSS, JW, JZ, and AS contributed to data extractions and cross-checking.

## Data summaries and analyses

We used the R computational environment v.4.3.1 [70] for final data cleaning, tabulating, analyses, and visualizations. These tasks were led by ML with feedback from SN. In brief, we summarized extracted award-level and individual winner-level characteristics, using data across all Subject Areas overall and additionally by subject area. Our data collection process was designed to minimize non-independence so only 1 eligible award was included per journal (the oldest award or the award with the highest cash reward). Further, in the included data

only a small proportion of awards (9%) were linked by the identity of an awarding society (18, where 13 societies were linked to 2 awards, 1 to 3 awards, 1 to 4 awards). If the same award was extracted for multiple Subject Areas, only 1 copy of the extracted data was retained to avoid pseudoreplication across Subject Areas (this resulted in reducing the final numbers of included and extracted awards in some Subject Areas below our initial target of 10).

For data visualizations, we counted frequencies of coded categorical data extraction responses, calculated proportions for each response option, and plotted key extracted variables as bar plots of counts or proportions overall and by subject area. We also explored the range of monetary benefits award winners receive by recalculating disclosed cash amounts into USD dollars using online exchange rates (GoogleFinance; https://www.google.com/finance/; July 2023) and plotted the histogram of the distribution of the values. For the binary information on the gender of past winners (female/male; we acknowledge that gender is not necessarily binary and that this approach potentially underestimates the prevalence of non-binary winners in the dataset), whether awardee profile was publicly posted (yes/no), whether awardee photo was publicly posted (yes/no). Following our preregistered plan, we plotted temporal trends across decades (years 2001 to 2010, 2011 to 2020, and 2021 to 2022). We made additional plots by year and fitted an overall logistic regression model with scaled award year as a predictor (presented in S1 Text). We visualized patterns in the genders and affiliation institutions and countries of the past winners. We noted the most commonly appearing institutions, countries, and first names. For countries, we visualized changes in the counts and proportions across decades and considered this pattern in relation to country scientific productivity (SCImago total document production per country in 2021; https://www.scimagojr.com/countryrank.php). Finally, we used a text mining approach to determine total word counts per award description and frequency of words in award descriptions providing additional context for award criteria and benefits. All data and code are openly available through a dedicated OSF project (https://osf.io/yzr7a/), Zenodo (https://doi.org/10.5281/zenodo.12465262), and GitHub repository (https://github.com/mlagisz/survey_best_paper_awards).

## Supporting information

**S1 Text. Methods and supplemental information.**
(HTML)

## Acknowledgments

We would like to thank Karim Khan (KK) and Jennifer Beaudry for their feedback and encouragement provided at the early stages of this project. The protocol for this project was developed during a hackathon event held during the AIMOS2022 conference and was also supported by the Open Life Science communities (OLS6 cohort).

## Author Contributions

**Conceptualization:** Malgorzata Lagisz, Joanna Rutkowska, Upama Aich, Robert M. Ross, Manuela S. Santana, Joshua Wang, Nina Trubanová, Matthew J. Page, Andrew Adrian Yu Pua, Yefeng Yang, Bawan Amin, April Robin Martinig, Adrian Barnett, Aswathi Surendran, Ju Zhang, David N. Borg, Jafsia Elisee, James G. Wrightson, Shinichi Nakagawa.

**Data curation:** Malgorzata Lagisz, Joanna Rutkowska, Upama Aich, Robert M. Ross, Manuela S. Santana, Joshua Wang, Nina Trubanová, Matthew J. Page, Andrew Adrian Yu Pua, Yefeng Yang, Bawan Amin, April Robin Martinig, Adrian Barnett, Aswathi Surendran, Ju Zhang, David N. Borg, Jafsia Elisee, James G. Wrightson, Shinichi Nakagawa.

**Formal analysis:** Malgorzata Lagisz.

**Funding acquisition:** Malgorzata Lagisz.

**Investigation:** Malgorzata Lagisz, Joanna Rutkowska, Upama Aich, Robert M. Ross, Manuela S. Santana, Joshua Wang, Nina Trubanová, Matthew J. Page, Andrew Adrian Yu Pua, Yefeng Yang, Bawan Amin, April Robin Martinig, Adrian Barnett, Aswathi Surendran, Ju Zhang, David N. Borg, Jafsia Elisee, James G. Wrightson, Shinichi Nakagawa.

**Methodology:** Malgorzata Lagisz, Shinichi Nakagawa.

**Project administration:** Malgorzata Lagisz.

**Resources:** Malgorzata Lagisz.

**Supervision:** Shinichi Nakagawa.

**Validation:** Malgorzata Lagisz, Joanna Rutkowska, Upama Aich, Robert M. Ross, Manuela S. Santana, Joshua Wang, Nina Trubanová, Matthew J. Page, Andrew Adrian Yu Pua, Yefeng Yang, Bawan Amin, April Robin Martinig, Adrian Barnett, Aswathi Surendran, Ju Zhang, Jafsia Elisee, James G. Wrightson, Shinichi Nakagawa.

**Visualization:** Malgorzata Lagisz, David N. Borg.

**Writing – original draft:** Malgorzata Lagisz.

**Writing – review & editing:** Malgorzata Lagisz, Joanna Rutkowska, Upama Aich, Robert M. Ross, Manuela S. Santana, Joshua Wang, Nina Trubanová, Matthew J. Page, Andrew Adrian Yu Pua, Yefeng Yang, Bawan Amin, April Robin Martinig, Adrian Barnett, Aswathi Surendran, Ju Zhang, David N. Borg, Jafsia Elisee, James G. Wrightson, Shinichi Nakagawa.

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
