## [Editor Report · Decision Letter 0]

28 Feb 2024

Dear Dr Lagisz, 

Thank you for submitting your manuscript entitled "Best Paper awards lack transparency, inclusivity, and support for Open Science" for consideration as a Meta-Research Article by PLOS Biology.

Your manuscript has now been evaluated by the PLOS Biology editorial staff, as well as by an academic editor with relevant expertise, and I'm writing to let you know that we would like to send your submission out for external peer review.

Once your full submission is complete, your paper will undergo a series of checks in preparation for peer review. After your manuscript has passed the checks it will be sent out for review. To provide the metadata for your submission, please Login to Editorial Manager (https://www.editorialmanager.com/pbiology) within two working days, i.e. by Mar 01 2024 11:59PM.

Kind regards,

Roli Roberts

Roland Roberts, PhD

Senior Editor

PLOS Biology

rroberts@plos.org

---

## [Decision Letter · Decision Letter 1]

1 May 2024

Dear Dr Lagisz,

Thank you for your patience while your manuscript "Best Paper awards lack transparency, inclusivity, and support for Open Science" went through peer-review at PLOS Biology. Your manuscript has now been evaluated by the PLOS Biology editors, an Academic Editor with relevant expertise, and by three independent reviewers.

You'll see that reviewer #1 is positive, but is not fully convinced by parts of the equity/EDI-related section, has some presentational suggestions, and wants you to tone down some of your assertions. Reviewer #2 is also positive, but finds the supplement to be unwieldy, recommending ways in which it could be improved. She also wants you to sharpen up your terminology and use it more carefully; she has a long list of textual questions. Reviewer #3 has a range of presentational requests, including shortening and focusing the Intro and Discussion; she also asks for clarification of several methodological points.

In light of the reviews, which you will find at the end of this email, we are pleased to offer you the opportunity to address the comments from the reviewers in a revision that we anticipate should not take you very long. We will then assess your revised manuscript and your response to the reviewers' comments with our Academic Editor aiming to avoid further rounds of peer-review, although might need to consult with the reviewers, depending on the nature of the revisions.

**IMPORTANT - SUBMITTING YOUR REVISION**

*Resubmission Checklist*

*Published Peer Review*

*PLOS Data Policy*

*Blot and Gel Data Policy*

Sincerely,

Roli Roberts

Roland Roberts, PhD

Senior Editor

PLOS Biology

rroberts@plos.org

REVIEWERS' COMMENTS:

Reviewer #1:

[identifies himself as Gerben ter Riet; see attachment for formatted version]

I enjoyed reviewing this manuscript. In my opinion it constitutes important and original work, clear objectives and using good methodologies, while drawing defensible and important conclusions that are supported by the data.

The paper shows that in a sample of 222 Best Paper awards across 27 disciplines a lot can be gained in terms of transparency of the organizational processes around these awards (opaque award criteria) and in terms of the required scientific features deemed important for submissions. For example, highlighting features such as outstanding and original, while rarely emphasizing e.g. the reproducibility or transparency of the work.

I am slightly less convinced by the equity part (EDI) of the paper ("to 150 evaluate potential gender and affiliation country disparities in the lists of past awardees."), which deals mainly with gender and country. I think this is because it is hard to exactly define from which point on some proportion proves that the mechanisms behind the awards are biased. For example, does the decrease in the proportion of award winning US papers after 2010 prove that the award processes were (more) biased between 2001 and 2010? In addition, the authors - for understandable reasons - refrained from assessing the quality (if that can be defined) of the award winning papers, hindering insights in whether the relative quality of US-based winners decreased after 2010 and may have caused the smaller proportion after 2010. The question of gender bias in research and science is complex and the authors may seriously consider to enrich their discussion by extending it to discussions around gender bias in grant application (e.g. https://bmjopen.bmj.com/content/10/8/e035058 ). After, all obtaining grants for many researchers is the foundation for writing research papers at all.

I agree with the authors that Best Paper awards have an important signaling function to (young) researchers on what good research is about. The lack of transparency and the emphasis on spectacular outcomes rather than reliable methods (process) is worrying. The current manuscript builds on earlier work by these authors (they also indicate that) in the field of ecology and evolution and extends it across 27 disciplines. 

The manuscript is well written and the figures are informative (I have 1 minor query in figure 1a: what does the vertical line of dots indicate? Add that to the legend). Importantly, the authors themselves preregistering their protocol. I was unable to access their sharing data and code (via osf.io). The paper would gain additional strength if data and code were shared. 

In their Abstract, the authors write that 'Instead, such awards increasingly rely on article-level impact metrics.' Since their work is based on a cross-sectional study, I wondered if they can truly claim the 'increasingly', which implies longitudinal data.

In their Introduction the authors loosely build an argument from an emphasis on striking results (rather than robust methods) and privileged marginalized groups. Although I understand their message, the line of argument from striking results to disadvantaged groups might be strengthened.

The Methods section is clear and informative. I wonder if the paper may shed some light (possibly hindered by small numbers) on the question whether awards endorsing EDI and/or open science features are more likely to be transparent in their award criteria.

In the Results section, how do the authors explain the rise in the number of Best Paper awards after 2010?

The Conclusions section reads more as a manifesto and although I am personally sympathetic to what they say, I wonder if these really can be counted as conclusions. A suggestion for the section title might be 'Further action'. The sentence 'Rectifying such biases is our collective responsibility.' assumes that there are biased, and there may be, but the data in my opinion do not really support this assumption. Extending the discussion section to the literature on biased in grant application procedures may be valuable.

Gerben ter Riet, MD PhD

Amsterdam, The Netherlands

Reviewer #2:

[identifies herself as Maia Salholz-Hillel; see attachment for formatted version]

Dear Editor,

Thank you for the opportunity to review this manuscript.

Overall, this paper makes an interesting and, to my knowledge, novel contribution to meta-research. Its focus on best paper awards brings attention to an incentive that has received less attention in the academic literature and ongoing research reform, and could serve to spark further discussion and change.

Furthermore, the authors "practices what they preach" with regards to open science: their study is pre-registered with a public protocol, and code and data is made available. Please note that due to time constraints, I could not review the protocol, code, or data in detail.

I had troubles interacting with the wonderfully detailed supplement and would suggest formatting changes: The PDF appears to be a printed Quarto-generated HTML, which is great for reproducibility; perhaps the HTML webpage (e.g., hosted on a GitHub Pages) could be linked within the PDF for better native navigation. The PDF pages are far larger than A4 and hard to use; I would suggest making these A4 and forcing page breaks for legibility. Additionally, the hyperlinks do not work, and with so many sections and so few pages, this made it a challenge to navigate; I imagine this is due to the peer review software and the final supplement would be "clickable." Also, it would be helpful to have numbered/lettered sections of the supplement, if journal policy allows, as it is extensive, and I found myself searching for the correct section. Additionally, due to time constraints, I was not able to review the supplement in detail.

The authors argue that best paper awards fall short in three interlinked areas: award criteria and details are not sufficiently explicit and clear to applicants and other scientists ("transparency"); awards make insufficient efforts to be inclusive of a broader range of scientists ("inclusivity"); awards give insufficient weight to responsible research practices highlighted in ongoing research reforms ("open science"). The first two ("transparency" and "inclusivity") regard how the award description is written or "award characteristics," per the authors; whereas "open science" regards the research itself. However, the biggest challenge I saw was the lack of clear delineation and connections between these dimensions; I felt the terminology somewhat was insuffiently defined and used too interchangeably, within the three areas, as well as with additional terminology (e.g., sometimes "inclusivity" and sometimes "diversity", though these are separate although related concepts). I believe that a careful pass of the language with some additional or re-writing as needed could help resolve these challenges. Below are some specific instances to help clarify what I mean:

- Title includes "transparency, ***inclusivity***, and support for Open Science," while abstract includes "transparency, ***diversity***, and openness in science" (emphasis added)

- As "research ***transparency***" is a term often used within ***open science*** (as on **lines 217, 272, 300**, etc.), it wasn't always clear whether "transparency" refered to the award (i.e., whether eligibility info is explicit) or the awarded research (i.e., whether the research is transparent about what was done; i.e., open science). Personally it would be clearest using another term, but as Lagisz and colleagues have previously published on this topic with this terminology, I would understand their desire for consistency, and, in that case, the use of the terms should be made clearer.

- In the discussion, the paragraph from **lines 292 - 317** starts with "Do Best Papers awards follow ***equitable*** practices or the path of least resistance?" (emphasis added) and then proceeds to discuss impact metrics and open science. However, is the issue at hand primarily one of inequity (open science practices may not always foster and may even extend inequity per Ross-Hellauer 2022 https://doi.org/10.1038/d41586-022-00724-0), or rather that the awards may not be comprehensively evaluating the "bestness" of papers by using simplistic measures (similar to what is argued in research reform)?

In their discussion, the authors argue that the demonstrated shortcomings of current best paper awards merit changes to the design of awards, and they provide concrete suggestions. I would be curious to learn how the authors think their findings might be taken into consideration with ongoing internation research evaluation reform (e.g., CoARA) and believe this would merit being addressed in the discussion beyond the brief reference on **line 299**.

Below are comments regarding specific lines:

**73-74**: The authors mention "global calls," which I then expected to be elaborated in the paper, but I found this just in a brief reference on **line 299**. Please make this link clearer.

**87-98**: The claim about the Matthew Effect and historically disadvantages and marginalized groups would be stronger with a reference, if available.

**90**: Do the authors mean "Studying awards" instead of "Awards"?

**92**: What "related institutions"? Those giving the awards, e.g., "granting institutions"?

**103**: What do the authors mean by "the ones most valued"?

**115**: I would suggest more specific language than "lack of transparency," as I first read this as open-science-related.

**121**: "Prizes" terminology is first introduced here. I had to re-reread since I first thought this was different than awards and instead were what the authors later call "perks." Perhaps establish these as synonyms, possibly even earlier.

**125**: What do the authors mean by "diversity of work"?

**134-137**: While I intuitively agree with this claim, I would suggest either citing or soften the language.

**138-139**: Are previous studies revealing opaque best paper award criteria limited to reference 33 in computer science? Please reword to clarify.

**146**: What does it mean for an award to "consider equity and inc

---

## [Editor Report · Decision Letter 2]

14 Jun 2024

Dear Dr Lagisz,

Thank you for your patience while we considered your revised manuscript "Best Paper awards lack transparency, inclusivity, and support for Open Science" for publication as a Meta-Research Article at PLOS Biology. This revised version of your manuscript has been evaluated by the PLOS Biology editors and the Academic Editor.

Based on our Academic Editor's assessment of your revision, we are likely to accept this manuscript for publication, provided you satisfactorily address the following data and other policy-related requests.

IMPORTANT - Please attend to the following:

a) Please could you adjust the Title very slightly to: 'Best Paper' awards lack transparency, inclusivity and support for Open Science [for clarity, note: Best Paper capitalised and in quotes; Oxford comma removed, retain capitals for Open Science]

b) I note that your code and data are in OSF, Zenodo and Github. Because Github depositions can be readily changed or deleted, please insure that all material that is in Github is also in Zenodo (especially the data and code needed to generate the main Figs).

c) Please cite the location of the data clearly in all relevant main Figure legends, e.g. “The data and code needed to generate this Figure can be found in https://doi.org/10.5281/zenodo.11215401"

d) I note that you thank the reviewers in the Acknowledgements ("We thank the three reviewers, Gerben ter Riet, Maia Salholz-Hillel, and Tracey Weissgerber, for their helpful comments and suggestions"). While we appreciate the sentiment, this is against PLOS policy, so please could you remove this?"

We expect to receive your revised manuscript within two weeks. 

*Published Peer Review History*

*Press*

Sincerely,

Roli Roberts

Roland Roberts, PhD

Senior Editor

rroberts@plos.org

PLOS Biology

CODE POLICY

DATA NOT SHOWN?

---

## [Editor Report · Decision Letter 3]

18 Jun 2024

Dear Dr Lagisz,

Thank you for the submission of your revised Meta-Research Article ""Best Paper" awards lack transparency, inclusivity and support for Open Science" for publication in PLOS Biology. On behalf of my colleagues and the Academic Editor, Ulrich Dirnagl, I'm pleased to say that we can in principle accept your manuscript for publication, provided you address any remaining formatting and reporting issues. These will be detailed in an email you should receive within 2-3 business days from our colleagues in the journal operations team; no action is required from you until then. Please note that we will not be able to formally accept your manuscript and schedule it for publication until you have completed any requested changes.

Sincerely, 

Roli Roberts

Senior Editor

PLOS Biology

rroberts@plos.org